# Detection of Urban Development in Uyo (Nigeria) Using Remote Sensing

**Etido Essien** [1,*] and **Samimi Cyrus** [1,2,3,*]

1   Climatology Research Group, University of Bayreuth, Zapf building 4, 95448 Bayreuth, Germany
2   Institute of African Studies, University Bayreuth, Klimatologie, 95440 Bayreuth, Germany
3   Bayreuth Centre of Ecology and Environmental Research (BAYCEER), University Bayreuth, Klimatologie, 95440 Bayreuth, Germany
*   Correspondence: Etido.Essien@uni-bayreuth.de (E.E.); Cyrus.Samimi@uni-bayreuth.de (S.C.)

**Abstract:** Uyo is one of the fastest-growing cities in Nigeria. In recent years, there has been a widespread change in land use, yet to date, there is no thorough mapping of vegetation change across the area. This study focuses on land use change, urban development, and the driving forces behind natural vegetation loss in Uyo. Based on time series Landsat Thematic Mapper (TM)/Enhanced Thematic Mapper Plus (ETM+)/Operational Land Imager (OLI) image data, the relationships between urban land development and its influencing factors from 1985 to 2018 were analyzed using remote sensing (RS) and time series data. The results show eight land use cover classes. Three of these (forest, swamp vegetation, and mixed vegetation) are related to natural vegetation, and three (sparse built-up, dense built-up, and borrow pit) are direct consequences of urban infrastructure development changes to the landscape. Swamp vegetation, mixed vegetation, and forest are the most affected land use classes. Thus, the rapid growth of infrastructure and industrial centers and the rural and urban mobility of labor have resulted in an increased growth of built-up land. Additionally, the growth pattern of built-up land in Uyo corresponds with socioeconomic interviews conducted in the area. Land use changes in Uyo could be attributed to changes in economic structure, urbanization through infrastructure development, and population growth. Normalized difference vegetation index (NDVI) analysis shows a trend of decreasing vegetation in Uyo, which suggests that changes in economic structure represent a key driver of vegetation loss. Furthermore, the implementation of scientific and national policies by government agencies directed at reducing the effects of urbanization growth should be strengthened, in order to calm the disagreement between urban developers and environmental managers and promote sustainable land use.

**Keywords:** Land use; remote sensing; urban expansion; Landsat; NDVI

## 1. Introduction

Land use change is a key component of research in environmental change and management worldwide [1–3]. In particular, urban growth and urban spread have drastically transformed the physical environment [4]. The most common change is the replacement of soil and vegetation with urban features such as concrete, asphalt, and buildings [4]. Consequently, this affects the albedo and runoff characteristics of the land layers and the environment, significantly affecting local and regional land–atmosphere energy exchange processes [4]. Hence, it is important to detect urban development in order to understand urban dynamics, ecology, and climatology, and to manage resources and services in urban environments [5–7]. In this context, urban ecologists try to understand how context changes the shape of urban systems, as they both drive and respond to environmental changes [8]. Therefore, understanding the interaction between urbanization and ecological processes is recognized

as a major task in urban ecology [9,10]. In today's world, the populations of big cities and clusters of cities are growing, as more people move to urban areas to participate in urban life, hoping to make a better living than in rural areas [11]. Accelerating urban growth has always been an indication of the vigor of a region's economy [5]. Such scattered growth, driven largely by technological advancement and population growth, has rarely been well planned, provoking concern over the degradation of environmental health [5]. It is essential to gain a better understanding of how urban dwellers are changing, especially in terms of how individuals add to and characterize urban growth and how urban growth can have consequences in changing the direction of population growth [12]. Even though the growth of mega-cities is often discussed, in Africa, major changes in urban populations normally happen in small and medium-sized cities [13].

In Nigeria, cities are struggling with rapid urban growth and development, and this plays a major role in land use and land cover changes in the country [14]. According to a study on land reform in Nigeria, land is seen as a social security asset to most people, because after all else has failed in the city, they can still manage to go back to their villages to demand a piece of their inherited family land and start subsistence farming [15]. Land control and land use policy must consider the fact that most people's existence relies on having access to a piece of land, and they would strongly resist and oppose any act to deprive them of this land [15]. Therefore, the government needs to pay more attention to land management and the planning of urban centers [16]. Based on detailed analysis, it is necessary to identify and evaluate land cover changes and urban settlement structures [16] in order to provide a judicious mapping of land use and urban development as a planning tool for Nigeria's economy [17].

Land use and land cover types are major indicators for understanding the relationship between environmental changes and human activities that occur in the environment [14]. Therefore, the detection of changes in land use reveals noticeable differences in the developmental process of a particular area, by monitoring it at different time periods [15]. It is an important aspect of research for detecting environmental changes [16]. Thus, in order to research past changes in land use and land cover in Nigeria, and to predict the future of land use changes, we need a proper understanding of the people who interact with the land and motivate anthropogenic activities and developmental processes [17].

The free availability of standardized multitemporal Landsat data [18] facilitates continuous analysis of temporal changes. Hence, Landsat has been used for mapping land surface changes and detecting disturbances (e.g., [19–22]), as well as for urban development (e.g., [23–31]). Two previous studies on Akwa Ibom State mainly focused on Uyo, though they did not capture the spatial development in the region and did not differentiate the built-up areas [32,33]. Their analysis was conducted for only two years (1986 and 2007). In this study, remote sensing data from Landsat Thematic Mapper (TM), Enhanced Thematic Mapper Plus (ETM+), and Operational Land Imager (OLI) sensors were used to analyze the temporal and spatial extent of urban area shifts and land surface changes in the region of Uyo (a medium-size city in southeastern Nigeria), by using historical data from Landsat from 1986 to 2017. The aim of this research is to analyze urban growth and its spatiotemporal differences in order to derive the driving factors of urban change. The overall aim is addressed by these three objectives:

(i)     To classify land cover classes for three years in order to quantify changes.
(ii)    To run a time series analysis to detect phases of urban growth, spatially and temporally.
(iii)   To link urban changes to the drivers of change.

The results are aimed at providing insight into the urban growth of a medium-size city (Uyo) in Nigeria to provide a basis for urban planning.

## 2. Materials and Methods

### 2.1. Study Area

This research was carried out in Uyo, the capital of Akwa Ibom State, Nigeria. The city lies between longitudes 37°50′ E and 37°51′ E, and between latitudes 55°40′ N and 54°59′ N. Uyo has a total

area of 188.035 km$^2$ with an estimated population of 3,920,208 [34]. Uyo is characterized by a tropical humid climate without distinct seasonal variation. It has an annual precipitation of almost 1000 mm, with only three months with considerably low rainfall, from December to February, and little variability in temperature [35]. It was a district headquarters during the colonial era and was later upgraded to a local government headquarters. In 1987, Uyo became the state capital of Akwa Ibom. The changes and status upgrade in the area attracted development. Figure 1 shows a map of the study area.

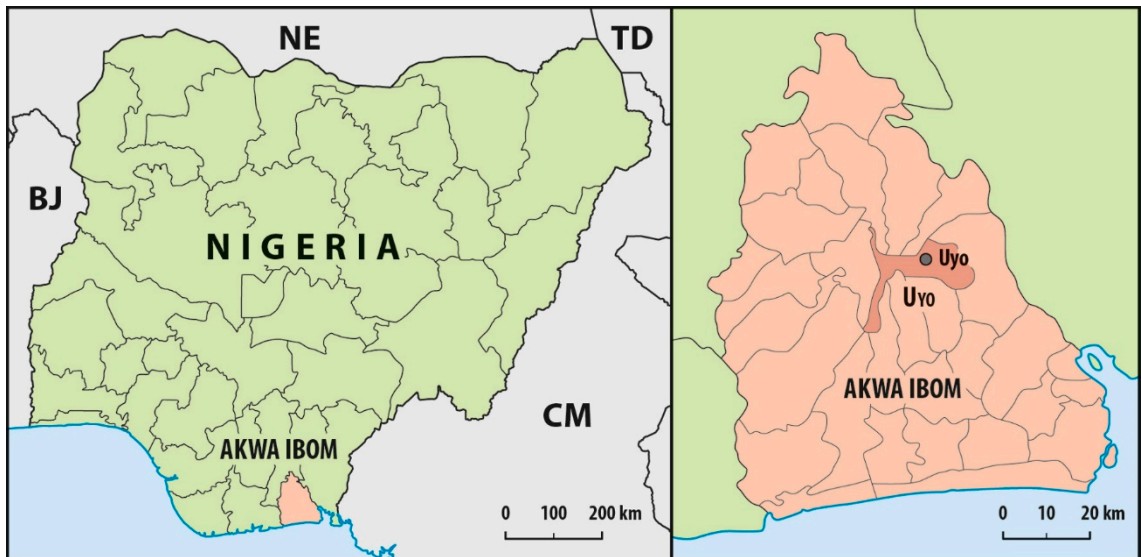

**Figure 1.** Map of Nigeria showing Akwa Ibom State, and map of Akwa Ibom State showing Uyo (Source: Ministry of Land Surveying, Uyo).

## 2.2. Data Collection

Global Positioning System (GPS) points were picked at different locations in Uyo, and information was also obtained from satellite images. During the field trip, different land cover types were identified, and photographs and coordinate points were taken at each location. Information on land use types in the area was gathered by interviewing native leaders who had worked in Uyo since it was established as a state capital, using their local languages. Based on the information obtained from the people interviewed and a reconnaissance survey conducted in the area, the land cover types of Uyo were classified into eight classes: forest, cropland, mixed vegetation, swamp vegetation, dense built-up, sparse built-up, borrow pit, and water bodies. Forest includes areas that are largely covered by trees and approximately more than 0.8 ha. Cropland includes areas that are mostly used for the cultivation of crops. Dense built-up includes areas that are populated with different kinds of buildings that are very close to each other (less than approximately 3 m). Sparse built-up includes areas that are scattered with both buildings and cropland. Mixed vegetation includes areas that comprise cropland, swamp vegetation, and forest land. Swamp vegetation includes bog areas with different tree covers. Borrow pit includes areas with large excavations of sand for construction work. Water bodies include areas characterized by flowing water, such as a lake, river, or stream [19].

## 2.3. Remote Sensing Data

Landsat images were used to monitor changes in land use and land cover from 1985 to 2018. Landsat Thematic Mapper (TM), Enhanced Thematic Mapper Plus (ETM+), and Operational Land Imager (OLI) images with cloud cover of less than 80% were bulk downloaded for the study area. Landsat (Tier 1) images are known to have the highest data quality, and they were radiometrically calibrated by the United States Geological Survey (USGS) when ordered using the standard method [36].

A total of 281 images, an average of eight Landsat images per year, were bulk downloaded. There were no Landsat images between 1992 and 1998 for the study area (path/row 188/56).

### 2.4. Training Samples and Reference Data

The training samples were chosen using an on-screen digitizing method in QGIS® [37], using red/green/blue (RGB) false color (urban) band combinations 7, 6, and 4 of Landsat images and other field data. Table 1 lists the numbers of training samples. For accuracy assessment, validation of an adequate amount of testing data is required per class of interest [38]. Therefore, it is essential to have sufficient testing data to test a statistically valid error matrix to represent classification accuracy. The validation of post–land cover classification results is naturally difficult due to the absence of field observations at different times [19]. In order to validate the accuracy of the 1986 and 2003 image classifications, the results were compared to the achieved 2017 images. Since there were validation data from the fieldwork and different classes in the post-classified images, it was still possible to identify some features.

**Table 1.** Samples of the training data.

| Category | Training Samples | |
| --- | --- | --- |
| | ROI | Pixels |
| Water bodies | 95 | 5321 |
| Swamp vegetation | 101 | 8942 |
| Sparse built-up | 105 | 10,422 |
| Mixed vegetation | 98 | 7356 |
| Forest | 97 | 8945 |
| Dense built-up | 94 | 8132 |
| Cropland | 110 | 12,435 |
| Borrow pit | 70 | 4851 |

ROI = region of interest.

### 2.5. Land Cover Classification and Change Detection

Changes in land cover from 1985 to 2017 were analyzed using a random forest (RF) algorithm. The accuracy of random forest has been generally affirmed in various fields, but it has not been used much in ecology [37]. Random forest is a classification that involves a combination of trees organized so that independent identical trees circulate random vectors and each tree casts a vote for similar trees in the same group [39]. Random forest also aids in learning complex nonlinear connections, such as those in various forest systems, and the connection of overstory to forest vegetation [40]. Textural information can give more knowledge to a random forest classifier by separating vegetation with similar spectral information [41]. Bagging seems to enhance accuracy and estimates the error rate of the trees when random features are used [39] and gives the importance of each tree in the classification. Accuracy and error rates are predicted for each tree using the bag predictions as well as the mean of all samples [42]. To extract the noise from image classification, we used a $9 \times 9$ majority cell filter for the different classes, and thereafter we mosaicked all the image tiles by finding the best position [41]. Following that, we used a nearest-neighbor method to restore pixels that had no data and performed an accuracy assessment to see if the image classification was a clear representation of Uyo land use. In order to reduce errors in land cover classification resulting from reflectance differences in the dry and rainy seasons [19], land cover classification was based on only rainy season Landsat images. Rainy season images have the advantage of showing crop cultivation and healthy vegetation [19]. The total accuracy of the land cover classification was evaluated using a confusion matrix to measure the similarity compared to 2017 classification results (Figure 2) and the field data. For individual

land cover types, a confusion matrix was created and the total accuracy, user and producer accuracy, and kappa statistic were calculated for every class type [19,37]. The total accuracy was determined by dividing the number of classified elements by the number of pixels added in the assessment levels [19]. The kappa statistic is another way of assessing classification accuracy by subtracting the effect of random accuracy and measures whether a classification is better than a random classification [19].

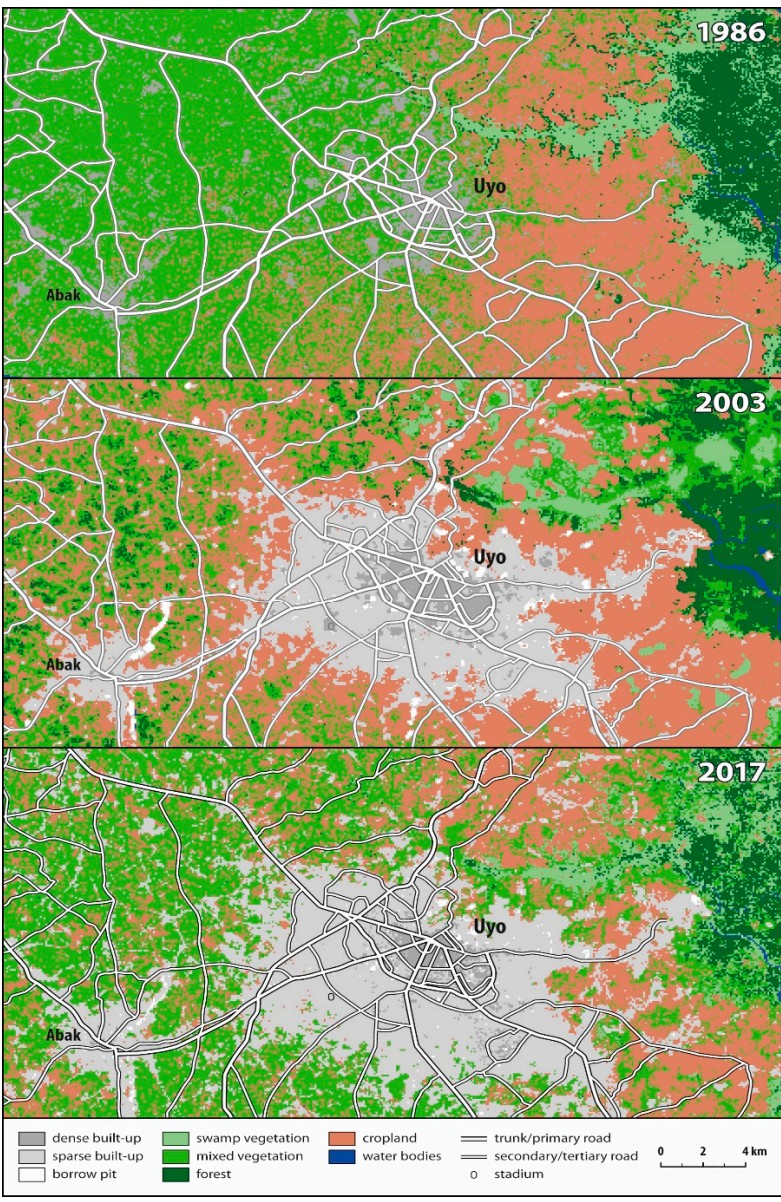

**Figure 2.** Land cover maps of Uyo for 1986, 2003, and 2017.

## *2.6. Time Series Analysis*

The Break For Additive Seasonal and Trend (BFAST) algorithm integrates the dissolution of time series into seasonal trends and remainder components in a way that can detect seasonal and abrupt changes within the trend. BFAST examines and provides functionality to detect a disturbance in real time and models it. The BFAST approach is very flexible and can detect missing data without interpolation [43]. Moreover, it has different models that can be used to fit the time series data and detect structural changes [43]. A total of 281 Landsat images were stacked in BFASTMONITOR; similarly, we also extracted the normalized difference vegetation index (NDVI) time series data from 1985 to 2013 (Figure 3) and created a map of changes that occurred in Uyo with time using the BFAST algorithm.

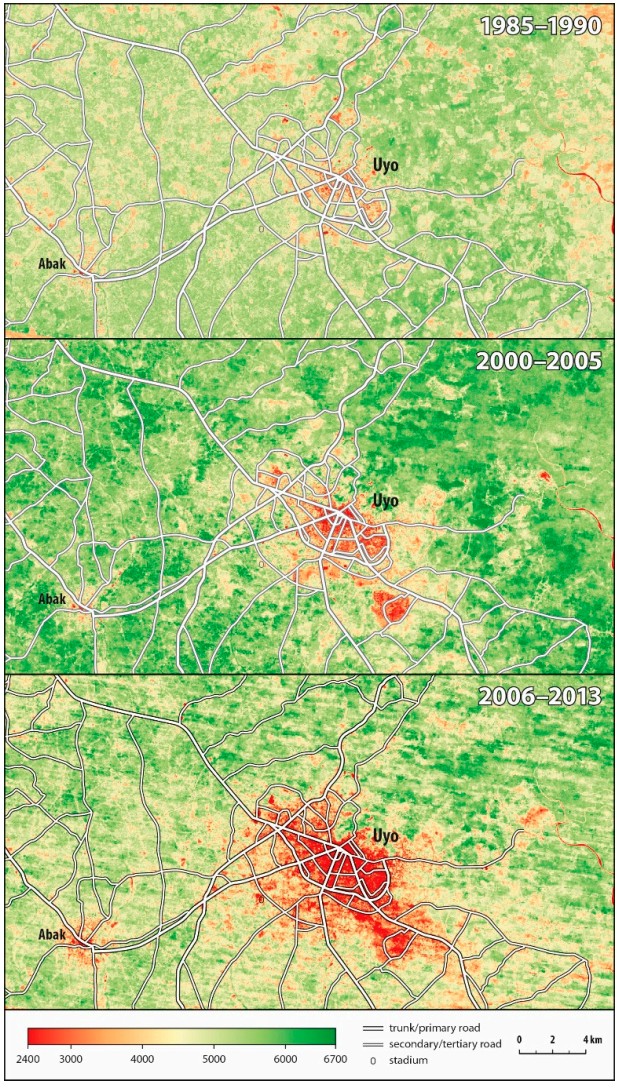

**Figure 3.** Median normalized difference vegetation index (NDVI) for the study area.

## 3. Results

### 3.1. Changes in Land Cover Patterns

To assess the quality of land cover classification, we used an internal out-of-bag random forest accuracy assessment method. Field and satellite data showed that the total accuracy of land cover change using a confusion matrix was 90% with a kappa statistic of 0.73 in 1986, 93% with a kappa statistic of 0.79 in 2003, and 95% with a kappa statistic of 0.83 in 2017 (Table 2). Land use and land cover types vary in Uyo; Figure 2 shows there was a trend of massive increase of built-up areas over the past 32 years. Table 3 shows there was a huge decline from 1986 to 2017 in forest area, from 414.94 to 219.41 km$^2$; swamp vegetation, from 48.94 to 18.31 km$^2$; and mixed vegetation, from 48.94 to 29.53 km$^2$. Changes in the built-up area grew from built-up land alone in 1986 to dense built-up, sparse built-up, and borrow pit in 2003, which have continued to grow. In 1986, built-up land had an area of 24.85 km$^2$; it increased to 15.75 km$^2$ dense built-up, 154.25 km$^2$ sparse built-up, and 4.85 km$^2$ borrow pit in 2003, and to 26.42 km$^2$ dense built-up, 251.51 km$^2$ sparse built-up, and 9.79 km$^2$ borrow pit in 2017. The total change in built-up area increased to 287.72 km$^2$ in 2017. This shows a significant increase in the built-up area with an average change of 8.98 km$^2$ per year, which means built-up land keeps changing every year due to the continuous increase in economic activity in the area.

**Table 2.** Landsat 1986, 2003, and 2017 image classification confusion matrix class.

(a) 1986

| | Built-Up | Cropland | Forest | Mixed Vegetation | Swamp Vegetation | Water Bodies | UA% |
|---|---|---|---|---|---|---|---|
| Built-up | **312** | 7 | 0 | 57 | 0 | 0 | 91% |
| Cropland | 1 | **439** | 0 | 23 | 0 | 0 | 86% |
| Forest | 0 | 0 | **440** | 0 | 40 | 1 | 82% |
| Mixed vegetation | 34 | 27 | 1 | **423** | 0 | 0 | 80% |
| Swamp vegetation | 0 | 0 | 49 | 0 | **383** | 0 | 92% |
| Water bodies | 0 | 0 | 8 | 0 | 0 | **173** | 96% |

Kappa statistic = 0.731, total accuracy = 90%, UA= user's accuracy.

(b) 2003

| | Borrow Pit | Cropland | Dense Built-Up | Forest | Mixed Vegetation | Sparse Built-Up | Swamp Vegetation | Water | UA% |
|---|---|---|---|---|---|---|---|---|---|
| **Borrow Pit** | **87** | 0 | 0 | 0 | 1 | 7 | 0 | 0 | 94% |
| Cropland | 0 | **404** | 0 | 0 | 6 | 0 | 0 | 0 | 96% |
| **Dense built-up** | 0 | 0 | **345** | 0 | 0 | 1 | 0 | 0 | 98% |
| Forest | 0 | 0 | 0 | **391** | 24 | 0 | 19 | 0 | 87% |
| Mixed vegetation | 0 | 14 | 0 | 20 | **366** | 0 | 10 | 0 | 89% |
| **Sparse built-up** | 3 | 0 | 0 | 0 | 0 | **438** | 0 | 0 | 98% |
| Swamp vegetation | 0 | 0 | 0 | 26 | 18 | 0 | **360** | 0 | 88% |
| Water bodies | 0 | 0 | 0 | 3 | 0 | 0 | 2 | **68** | 98% |

Kappa statistic = 0.792, total accuracy = 93%, UA= user's accuracy.

(c) 2017

| | Borrow Pit | Cropland | Dense Built-Up | Forest | Mixed Vegetation | Sparse Built-Up | Swamp Vegetation | Water Bodies | UA% |
|---|---|---|---|---|---|---|---|---|---|
| **Borrow Pit** | **154** | 1 | 0 | 0 | 0 | 2 | 0 | 0 | 97% |
| Cropland | 0 | **336** | 0 | 0 | 0 | 0 | 0 | 0 | 98% |
| **Dense built-up** | 0 | 0 | **212** | 0 | 0 | 1 | 0 | 0 | 100% |
| Forest | 0 | 0 | 0 | **369** | 3 | 0 | 0 | 0 | 93% |
| Mixed vegetation | 0 | 0 | 0 | 3 | **270** | 0 | 1 | 0 | 93% |
| **Sparse built-up** | 1 | 0 | 0 | 0 | 0 | **379** | 0 | 0 | 95% |
| Swamp vegetation | 0 | 0 | 0 | 0 | 3 | 0 | **313** | 0 | 97% |
| Water bodies | 0 | 0 | 0 | 3 | 0 | 0 | 0 | **98** | 100% |

Kappa statistic = 0.839, total accuracy = 95%, UA= user's accuracy; Numbers on the diagonal of the matrix show trees that vote in each class (identified in bold), off-diagonal numbers are misclassification (error of commission).

**Table 3.** Land cover use statistics of Uyo from 1986 to 2017.

| Land Cover Class | 1986 Land Cover (km²) | 2013 Land Cover (km²) | 2017 Land Cover (km²) | Change in Land Cover (km²) 2003–2017 |
|---|---|---|---|---|
| Built-up | 24.85 | | | |
| Dense built-up | | 15.75 | 26.42 | 10.67 |
| Sparse built-up | | 154.23 | 251.51 | 97.26 |
| Borrow pit | | 4.85 | 9.79 | 4.94 |
| Swamp vegetation | 48.94 | 28.91 | 18.31 | −10.6 |
| Mixed vegetation | 44.2 | 35.38 | 29.53 | −5.85 |
| Cropland | 232.73 | 261.6 | 284.41 | −19.81 |
| Forest | 414.94 | 249.35 | 219.41 | −29.94 |
| Water bodies | 1.01 | 1.04 | 1.01 | |

### 3.2. NDVI Trends in Uyo

NDVI time series (Figure 3) show that from 1985 to 1990, the changes in vegetation in Uyo occurred at a slower rate until 2000 and beyond, when the vegetation (forest, swamp vegetation, mixed vegetation) started decreasing rapidly without increasing any time of the year, while the cropland went up and down because of its seasonal variation (Figure 4). Applying BFAST to the NDVI time series for urban growth generated over time shows significant changes in the time trend component. The changes were detected by creating a loop and extracting the average pixels in the NDVI data of the study area (Figure 4). This also shows the yearly trend (Yt) of vegetation (forest, swamp vegetation, mixed vegetation) in Uyo, which indicates that it keeps decreasing without recovering in any year, and the time trend (Tt) shows a continuous trend of decreased vegetation in Uyo due to a continuous increase in built-up area, while the seasonal trend (St) shows changes in cropland continuing at a normal rate.

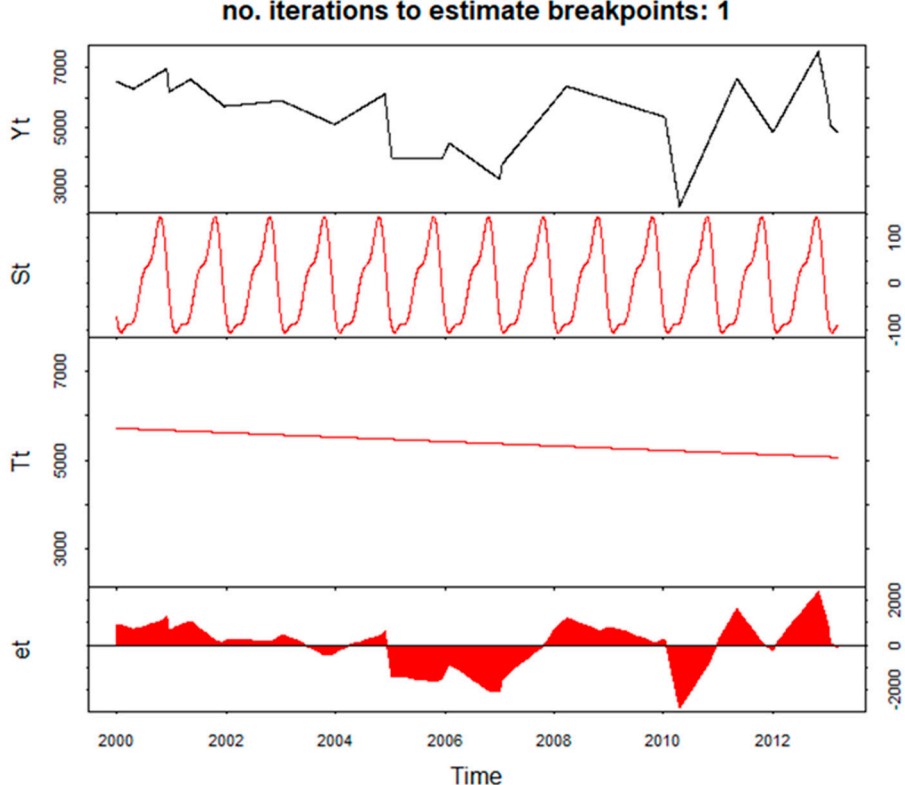

**Figure 4.** Yearly time (Yt), seasonal trend (St), time trend (Tt), residual component (Et).

*3.3. Changes in Economic Structure*

The achievements in economic development of the state can be evaluated in terms of the income received from the federal account, which was increased from 8 billion naira to 17.5 billion naira in 2007, since Akwa Ibom State is one of the Niger Delta states that produce crude oil, a major export of the country. Growth in the economy boosts the process of urban construction, with a large demand for land resources increasing economic activities; likewise, the economic output could lead to increased land demand for urban expansion [44]. With robust economic growth, the local government of Uyo, the capital of Akwa Ibom State, aiming to expand its internally generated revenue for development purposes, demolished thatched houses and shop shade covers and even relocated a big market in the city center that was in the dense built-up map of 2003 (Figure 2), in order to construct modern roads and create drainage channels for flood control in the city. This shows that economic growth triggered a need for more land development, leading to rapid growth of built-up areas from the urban center to its surrounding areas (Figure 2), as seen on the 2017 map.

*3.4. Population Growth*

The National Census of Nigeria carried out in 1991 put the provisional population of Akwa Ibom State at 2,409,613. Out of this figure, the population of urban Uyo was 244,762 [34]. The current population of the state based on the published National Population Commission (NPC) census record is 3,920,208. In 2006, urban Uyo alone was 309,573 [34]. As a result of the strategic location of the town and its being the state capital, there is a corresponding significant influx of people from far and near. Apparently, this is directly responsible for the built-up expansion in urban Uyo (Figure 2). Another effect of population explosion, especially in developing countries like Nigeria, is increased poverty levels among the people due to a lack of sufficient resources for the growing populace [45]. This could be a precedent for overcrowding of urban cities, wrong use of land and environmental resources, and consequently huge loss of biodiversity.

## 4. Discussion

Urban growth in developing nations is mostly caused by population growth due to infrastructure in urban centers and the mobility of labor [46]. The results of this research are in agreement with previous studies [32,33], but there is much improvement in our study. The area of our research was not centered around urban Uyo alone but included Uyo and its surroundings. Also, the direction of change is shown in our study (Figure 2). Moreover, there are also new findings in this research because of ongoing development in the area. The urbanization of Uyo, which before was centered around the city center, what is today designated as the Ibom Plaza, has expanded to remote areas of Mbiabong, Etoi, Afaha Oku, Ikot Oku Ubo, Ofot, Osong Ama, Itam, and Mbierebe [32]. This has now shifted direction to the Abak local government area and its surroundings. Figure 2 shows that there is densification in the center of Uyo and there is no expansion toward the northern part of the area, meaning that the development trends mainly follow the road network in Uyo. Changes in raster always result in breakpoint timing values, so we have to convert breakpoints into yearly changes in the raster layers (Figure 5). The highlighted color in the maps in Figure 5 shows yearly changes that occurred in Uyo for some monitored pixels at different times of the year. There was almost no change before 2010 and not much change after 2013. Major changes took place in 2010 and 2011, and this was the peak period when the government cleared a lot of land for development purposes, since its aim was to transform Uyo into a modern city (Figure 5). The interviews conducted in the area show that land is cheaper in the local Abak area and it has a direct access road to Uyo (Figure 5), so average people can easily acquire land in the local Abak area.

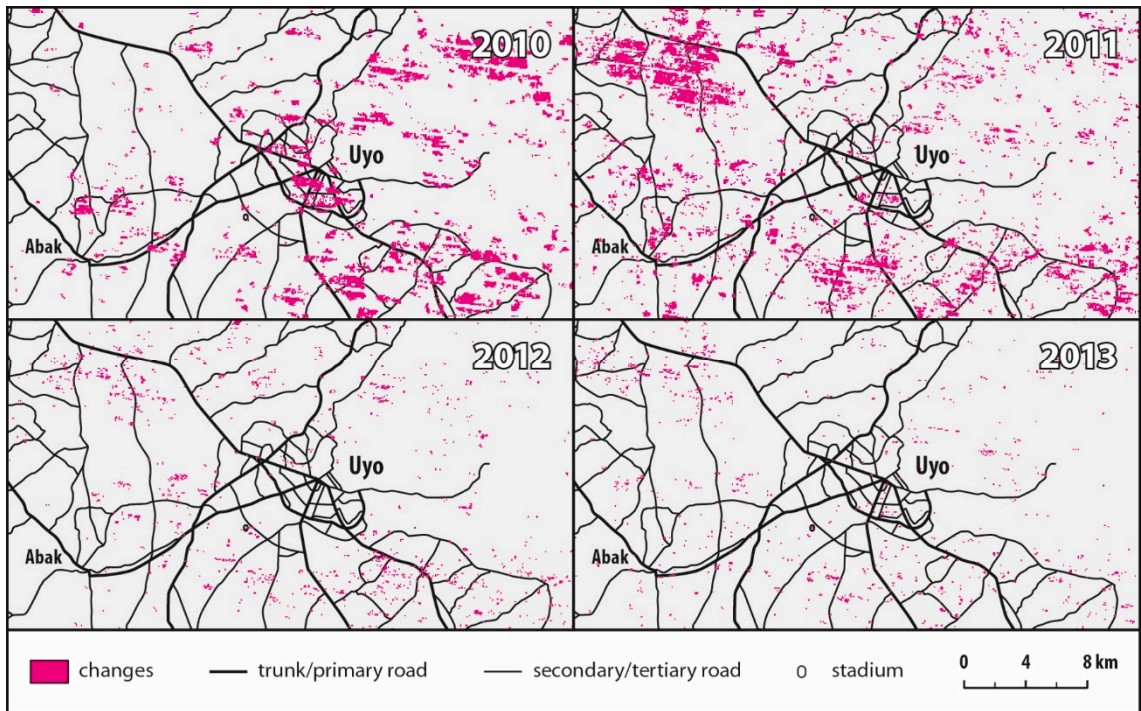

**Figure 5.** Time series data of Uyo showing changes over monitored years in the study area.

Population statistics of Uyo show that the effect of population growth in the urban areas is faster, while it is slower in the rural areas, because out of the total population of Akwa Ibom State, 78% of the people stay in Uyo. Similarly, other studies on urban development have also shown that population growth is one of the major factors of urban growth in Uyo. Again, with the recent robust allocations from the federal government to the state government, the government tries to make judicious use of the money by investing more in infrastructure development in the city. Thus, Uyo, as the capital city, has attracted several government institutions. The results of this research also show a dramatic shift in land cover change, which corresponds well with changes in structures of land surface and land use in Uyo. This research also shows the direction of land cover change in Uyo, which is moving from the center of the city to the local Abak area (Figure 2). The analysis further shows details of the spatial characteristics of changes in the built-up area in Uyo from the slow development period to the social economic development period from 1985 to 2017 (Figure 2). In general, the increased urban built-up area was caused by changes in economic structure, population growth, and urbanization. During this rapid economic development period, there was an obvious trend of population migration from rural areas to urban regions mostly for the mobility of labor, and this is considered to be one of the important driving forces. The classification accuracy of 95% shows that satellite data were efficiently used to analyze different types of vegetation in Uyo. Further analysis of the results from interviews conducted in the area shows that changes in economic structure, urbanization, and population growth are the most important causes of expansion of the urban built-up area within a limited time scale of 17 years.

*Urbanization*

Urban expansion is indicated by increased population in an area. The results of the census conducted in 1991 and 2006 in Uyo show a general increase in the populations of urban centers [34]. Urban growth has tremendously increased the urban population and urbanized some nearby rural areas. Consequently, this has resulted in various impacts on the environmental systems on a larger scale [43,47,48]. Another major impact of urbanization is deforestation and changes in land cover/land use. Landsat images of the study area for 1986, 2003, and 2017 clearly explain this effect. The areas of vegetation cover show a drastic decrease. This change can be attributed to new forms of land use,

such as infrastructure development. The previous administration constructed a modern stadium with a capacity of 40,000, currently said to be the best stadium in Nigeria, which is frequently used by the national football team. Additionally, there are newly built structures such as Silver Bird Cinema Centre, a five-star hotel with a modern golf resort, newly constructed roads, an E-library, and Ibom International Airport. In fact, all these were built in the last eight years. Figure 2 shows that these new structures have opened new links and development directions toward the Abak area. Population growth has led to the development of houses by the state government in and around the closely settled zones of urban Uyo to meet the high demand. The state is said to surpass most other states in terms of housing infrastructure delivery [45,49]. The socioeconomic interviews conducted in the area show that investors tend to acquire more land than public workers, and community people are willing to sell their land because the investors are thriving in their businesses and usually purchase the land at high prices. This shows that development triggers the public to invest more in infrastructure in Uyo, and there is a continuous stretch of agricultural land lost to urban growth. Moreover, it shows that there was an intense need for land development due to rural–urban migration and labor mobility during the recent industrialization and urbanization, and this encouraged excessive land use in Uyo and stimulated the growth of built-up lands.

## 5. Conclusions

All three objectives were achieved in this research. It was possible to perform a robust classification showing a differentiated pattern of land cover change and temporal changes. The results show significant land cover change in the mixed vegetation and forest areas. These areas are close to the center of Uyo for flood control. This was analyzed by using satellite images and running a time series analysis on them. The main observed land cover change trends occurred after 2000, mainly from forest to cropland and sparse built-up. Although the BFAST method to detect changes in vegetation over a long period was appropriate, it could still be improved because it cannot fully detect missing data over a long period. Further research will be needed to discuss this problem. The changes correspond with increasing economic and political decisions combined with poor governance, leading to population growth and unplanned urban development. The master plan of Uyo keeps changing because of infrastructure development by different governments, and this directly affects the unstable land cover change. Therefore, to reestablish Uyo as a state with a secure environment, it is necessary to ensure socioeconomically and environmentally sustainable development. For more insights on differentiated urban development, socioeconomic data should be linked to satellite data to evaluate how household income directly affects urban development and differentiation of urban patterns.

**Author Contributions:** Conceptualization, E.E.; methodology, E.E.; software, E.E.; validation, E.E.; formal analysis, E.E.; investigation, E.E.; writing—original draft preparation, E.E.; writing—review and editing, S.C.; supervision, S.C.

**Funding:** This research was funded by German Academic Exchange Service (DAAD), grant number 91666076.

**Conflicts of Interest:** The authors declare no conflict of interest.

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
