# Peer review of "Detection of Urban Development in Uyo (Nigeria) Using Remote Sensing"

_land, doi:10.3390/land8060102_

Round 1

Reviewer 1 Report

1. It is not easy to understand the whole aspects of the study. It is important to write clear the objectives and conclusions.

2. In the method section important information about the used methods are missing.

3. Image processing includes radiometric and atmospheric corrections; what techniques were exactly performed? With what results?

4. Only six classes have been adopted for an area of ~ 188 km2, for a spatial resolution of 30m this is quite strange. This part must be robustly defended.

5. Supervised classification was performed, but no description of classes separability is given. Which instead is important to understand whether the classes chosen are good enough to portrait the variety of landcover of the region and whether they are easy to distinguish or not.

In conclusion, I am confident that authors performed a valuable work producing and interesting case study, however, given the many caveats of this paper several parts must be rewritten considering all the comments.

Author Response

Response to Reviewer 1 Comments

Point 1: It is not easy to understand the whole aspects of the study. It is important to write clear the objectives and conclusions.

Response 1: The correction has been made.

Point 2: In the method section important information about the used methods are missing.

Response 2: The correction has been made.

Point 3: Image processing includes radiometric and atmospheric corrections; what techniques were exactly performed? With what results?

Response 3: Landsat images ordered, were atmospherically corrected by USGS.

Point 4: Only six classes have been adopted for an area of ~ 188 km2, for a spatial resolution of 30m this is quite strange. This part must be robustly defended.

Response 4: Our aim was not to give land use classes in Uyo, so we focused on land use classes that were related to urban growth. During the field survey, land use classes in the study area where closely examined before we classified them into eight classes not six. The eight classes are: forest, cropland, mixed vegetation, swamp vegetation, dense built-up, sparse built-up, borrow pit and water bodies.

Point 5: Supervised classification was performed, but no description of classes separability is given. Which instead is important to understand whether the classes chosen are good enough to portrait the variety of landcover of the region and whether they are easy to distinguish or not.

Response 5: We performed a random forest classification not supervised classification. Forest land are areas that are largely covered by trees and approximately more than 0.8 ha. Cropland are areas that are mostly used for cultivation of crops. Dense built-up are areas that are populated with different kinds of building that are very close to each other approximately less than 3m space. Spares built-up are areas that are scattered with both buildings and cropland. Mixed vegetation are areas that comprises of cropland, swamp vegetation and forest land. Swamp vegetation are bog areas that has different trees cover. Borrow pit are areas with large excavation of sand for construction work. Water bodies are areas characterize by flowing water, example lake, river or stream [19].

Reviewer 2 Report

I have seen the corrections made and now the work is clearer.
Some image captions go to the following pages. Where possible, place them on the same page.

# Line 257 urban growth ... -> Urban growth

Now I believe that the work is complete and understandable even in the results shown.

Author Response

Response to Reviewer 2 Comments

Point 1: Line 257 urban growth ... -> Urban growth

Response 1: The correction has been made.

Round 2

Reviewer 1 Report

Thank you for revising this manuscript. The revised version is a much clearer presentation of the research you have conducted.
Many of the comments made in the previous review have been satisfactorily addressed.

Line 11: Landsat TM/ETM/OLI+ or Landsat TM/ETM+/OLI?

Line 118: Enhanced Thematic Mapper Plus (ETM+)

line 120: were atmospherically corrected by USGC  when order [19]. explain the method, it is important.

References: Please use one style for all references.

Author Response

Response to Reviewer 1 Comments

Point 1: Line 11: Landsat TM/ETM/OLI+ or Landsat TM/ETM+/OLI?

Response 1: it’s Landsat TM/ETM+/OLI. The correction has been made.

Point 2: Line 118: Enhanced Thematic Mapper Plus (ETM+) 

Response 2: The correction has been made.

Point 3: line 120: were atmospherically corrected by USGC when order [19]. explain the method, it is important.

Response 3: Landsat (Tier1) data is known to have the highest data quality was radiometrically calibrated by USGS when ordered using the standard method [36].

Point 4: References: Please use one style for all references.

Response 4: The correction has been made.

This manuscript is a resubmission of an earlier submission. The following is a list of the peer review reports and author responses from that submission.

Round 1

Reviewer 1 Report

In this manuscript, the authors report an analysis of land cover and land use change in Uyo based on classification of long-term Landsat imagery.  However, the experimental work is not thorough or precise and without appropriate controls, and the manuscript is written with poor English.  In my opinion, it cannot be accepted as a scientific paper.

1. As stated in line 126-127, the important purpose of this study is to identify the driving forces of the land use and landcover change, but all the results presented are all about the changes in land classes and NDVI, nothing about the changes in the potential driving forces, e.g. population, economics, policies…. in Line 307-308, the authors address the urban growth is mostly caused by population growth. How did the authors get this conclusion ?

2.  The results are based on random forest classification of Landsat data.But there is no validation of the classification results.  Also, the training sample size is relatively small comparing the whole study area.  It is well-known that the spectral  properties of vegetated surfaces (forest, cropland…) have dramatic seasonal variations.  How did the authors deal with the seasonal issue?

3. Table 2 only shows the classification accuracy of 2017. What about the other years? Also, table 2 is very hard to interpret,  I suggest the authors re-do all the tables and figures. 

4. According to Fig.2, the cropland increased from 2003 to 2017 and urban area decreased. What are the reasons behind it?  On the right side of the figures, no water body in 1986 and 2003, but there is a big water body in 2017. Why is that?What is the season of the Figure 2?

5. The result of figure 3 does not agree with figure 2.  Figure 3 shows  an on-going  decrease trend of vegetation from  1985 to 2013.  However, figure 2 shows an increase in cropland from 2003 to 2017.   I suggest the authors provide  good explanations.  

Reviewer 2 Report

It is not easy to understand the whole aspects of the study. It is important to write clear the introduction, objectives, results and conclusions.

According to this reason, my evaluation to this manuscript is negative.

Reviewer 3 Report

# line 102-105 remove dot after '....[35-36,  5237]'
--> Remote sensing   has   immense   contributions   in   biodiversity,   conservation   planning, deforestation   rates,   habitat   degradation,   and   shows   areas   in   human   altered 104landscapes  [35-36,  5237] and maybe  a  practical  tool  to  monitor  urban  growth.

# line 114
analysing areas such as Uyo. this study, remote sensing data from theLandsat TM,
--> This study....

2. Materials and Methods
# line 140
 (Figure 1) shows the map of the study area. -->  Figure 1 shows the map of the study area.
...the same is on line 164 with (Table 1)....

# line 153
Figure: 1 --> Figure 1.

2.4. Training Sample and Reference Data
You didn't explain the data used to define the ROI using on screen digitisation methods.For example you used the composition RGB,false color Landsat images and other data.

# line 165-166 I would write better this part.
[48] hows that  it  is  of  essence to  have  sufficient testing  data  for  testing a  valid statistically error matrix torepresent classification accuracy.

2.5?????
2.5 Land Cover Classification and Change Detection OR 2.5.Time Series Analysis ?

# line 218
statistic of 0.912(Table 2). the ....  --> statistic of 0.912(Table 2). The

and

"accessing landcover classaccuracy using confusion matrix,field and satellite data showed that the accuracy of land coverchange analysis was 0.89% error and a Kappa statistic of 0.912(Table 2). the accuracies error range from 0.02, %, 0.00%, 0.01%."  
I did not understand this part well. Can you explain it better?

# Page 9.
Pag.9 There is a blank space from lines 231...to lines...237.

# table 2
It is not well formatted.It is not clear

In general the paper is not clear to me and I think you  should revise the text formatting. All the parts of the text are not clear.